# Phasic activation of ventral tegmental neurons increases response and pattern similarity in prefrontal cortex neurons

**Motoko Iwashita[1]***

[1]National Institute of Mental Health, National Institutes of Health, Bethesda, United States

**Abstract** Dopamine is critical for higher neural processes and modifying the activity of the prefrontal cortex (PFC). However, the mechanism of dopamine contribution to the modification of neural representation is unclear. Using in vivo two-photon population $Ca^{2+}$ imaging in awake mice, this study investigated how neural representation of visual input to PFC neurons is regulated by dopamine. Phasic stimulation of dopaminergic neurons in the ventral tegmental area (VTA) evoked prolonged $Ca^{2+}$ transients, lasting ~30 s in layer 2/3 neurons of the PFC, which are regulated by a dopamine D1 receptor-dependent pathway. Furthermore, only a conditioning protocol with visual sensory input applied 0.5 s before the VTA dopaminergic input could evoke enhanced $Ca^{2+}$ transients and increased pattern similarity (or establish a neural representation) of PFC neurons to the same sensory input. By increasing both the level of neuronal response and pattern similarity, dopaminergic input may establish robust and reliable cortical representation.

## Introduction

The prefrontal cortex (PFC) plays an important role in adaptive behavior such as associative learning (*Duncan, 2001*). Dopaminergic input from the ventral tegmental area (VTA) is crucial for PFC function (*Schultz, 2007*). In primates, sensory cues, which are used in associative learning tasks, create specific temporal activity patterns in PFC neurons and a neural representation of the sensory cue (*Jacob et al., 2013*). However, how dopamine contributes to form neural representations at the neuronal network level is largely unknown. One potential mechanism is that dopaminergic (DA) neurons target dopamine over a number of inhibitory and excitatory neurons via their widespread axonal arborizations (*Matsuda et al., 2009*). Thus, investigating modification of neuronal activity at the population level will reveal the role of dopamine signaling in the formation of neural representation. Therefore, utilizing in vivo two-photon $Ca^{2+}$ imaging in awake mice, this study investigated how neural representation in PFC neurons is developed under regulation by dopamine, in response to visual sensory input.

## Results and discussion

PFC neuronal activity was recorded through a cranial window at the secondary motor cortex (M2). The M2 is categorized as the dorsomedial PFC in rodents in some publications, based on both its anatomical features, including thalamocortical and cortical–basal ganglia connections, and its functional role in motor decision (*Sul et al., 2011*; *Uylings et al., 2003*; *Hoover and Vertes, 2007*). The M2 receives inputs from both the VTA and the secondary visual cortex lateral area (V2L) (*Figure 1A,B*). Microelectrodes were implanted in these two brain areas to supply electrical stimulation (*Figure 1C,D*).

To determine the single input response, the PFC neuronal response to VTA stimulation was first measured by recording $Ca^{2+}$ transients. Most DA neurons are spontaneously active with firing patterns that range from regular tonic firing (1–5 Hz) to phasic firing (40–50 Hz) (*Robinson et al., 2004*; *Grace and Bunney, 1984a*; *Grace and Bunney, 1984b*). In addition, salient events such as

***For correspondence:**
moko0927@gmail.com

**Competing interests:** The author declares that no competing interests exist.

**Reviewing editor**: Michael Häusser, University College London, United Kingdom

**eLife digest** Around 120 years ago, Ivan Pavlov unintentionally sparked a new field of psychology research. He did so by noting that his dogs had learned to associate the sound of the bell that he rang before feeding them with the food itself, such that they would salivate upon hearing the bell even when there was no food present. This form of learning—now known as associative learning—has since been demonstrated in species from honeybees to humans.

For the brain to associate two events, such as the sound of a bell and the delivery of food, it must encode the first event and keep that information available or 'on-line' until the occurrence of the second event, at which point the two can be linked together. This process takes place in part of the brain called the prefrontal cortex, but the mechanism by which it occurs is largely unclear.

Now, Iwashita has obtained new insights into the molecular basis of associative learning by studying how the activity of the prefrontal cortex is affected by the activity of a second region of the brain. This second region, called the ventral tegmental area, is part of the brain's reward circuit: it becomes active whenever an animal experiences a desirable event, such as receiving food, and supplies a neurotransmitter called dopamine to its target areas, which include the prefrontal cortex.

Electrodes were used to mimic the changes in brain activity that occur when a mouse learns to associate a visual stimulus with a reward: this involved repeatedly activating the visual cortex in a conscious mouse, followed by activation of the ventral tegmental area. Short-lived increases in calcium levels were seen in the prefrontal cortex, raising the possibility that these 'calcium transients' are the signal that enables the brain to link two events. Moreover, blocking proteins called dopamine D1 receptors in the prefrontal cortex reduced the calcium transients, which is consistent with existing evidence that dopamine from the ventral tegmental area is required for associative learning.

Intriguingly, the calcium transients lasted for roughly 30 s, which is also the maximum length of time by which a stimulus and a reward can be separated and still be associated. Given that the calcium transients could not be detected in anesthetized mice, a full understanding of the mechanisms underlying associative learning may require studies of the conscious brain.

experiencing a novel environment or receiving a reward-associated signal, evoke phasic firing in the VTA (*Schultz, 2007*; *Horvitz, 2000*). The PFC activity evoked by VTA stimulation in these physiological ranges was therefore examined.

When low frequency VTA microstimulation was used (1 Hz–10 Hz, tonic range), $Ca^{2+}$ transients in the PFC neurons were evoked and decayed within 5 s. This steep increase and short decay is comparable with the reported response to sensory input observed in cortical neurons of anesthetized animals (*Ohki et al., 2005*). In contrast, high frequency VTA stimulation (40–50 Hz, phasic range) evoked substantially elongated $Ca^{2+}$ transients that lasted 20–30 s and then returned to the original baseline (*Figure 2*). In addition, these $Ca^{2+}$ transients of the PFC neurons reached a peak relatively slowly, 6–7 s after stimulation. Furthermore, high frequency VTA stimulation robustly evoked $Ca^{2+}$ transients in all tested animals, whereas low frequency VTA stimulation did not as only about half of the animals tested showed detectable $Ca^{2+}$ transients in the PFC. Importantly, these long-lasting $Ca^{2+}$ transients were only detected in awake and not in anesthetized mice (*Figure 1E,F*), possibly because of the potential inhibition of voltage-gated calcium channels by isoflurane (*Herring et al., 2009*; *Study, 1994*). This result highlights the advantage of a system using awake animals to elucidate dopamine regulation of neuronal responses in the PFC.

To test whether long-lasing $Ca^{2+}$ transients can be evoked by other neural inputs, the same microstimulation protocol used in the VTA (1–50 Hz) was applied to a region of the visual cortex, the V2L. Both stimulation frequencies only evoked short $Ca^{2+}$ transients in PFC neurons (*Figure 2—figure supplement 1*). Input from the VTA, especially through high frequency, phasic stimulation can therefore induce a specific signal required to evoke long-lasting $Ca^{2+}$ transients in PFC neurons. Phasic stimulation of the VTA is known to facilitate DA release from its terminals (*Tsai et al., 2009*; *Lavin et al., 2005*), therefore, it was hypothesized that DA receptors are involved in induction of the long-lasting $Ca^{2+}$ transients. DA receptors are members of the G-protein coupled receptor family and are composed of two groups: D1 and D2, which have opposing effects on intracellular signaling (*Seamans and Yang, 2004*; *Soltani et al., 2013*). Selective DA receptor antagonists for each family were

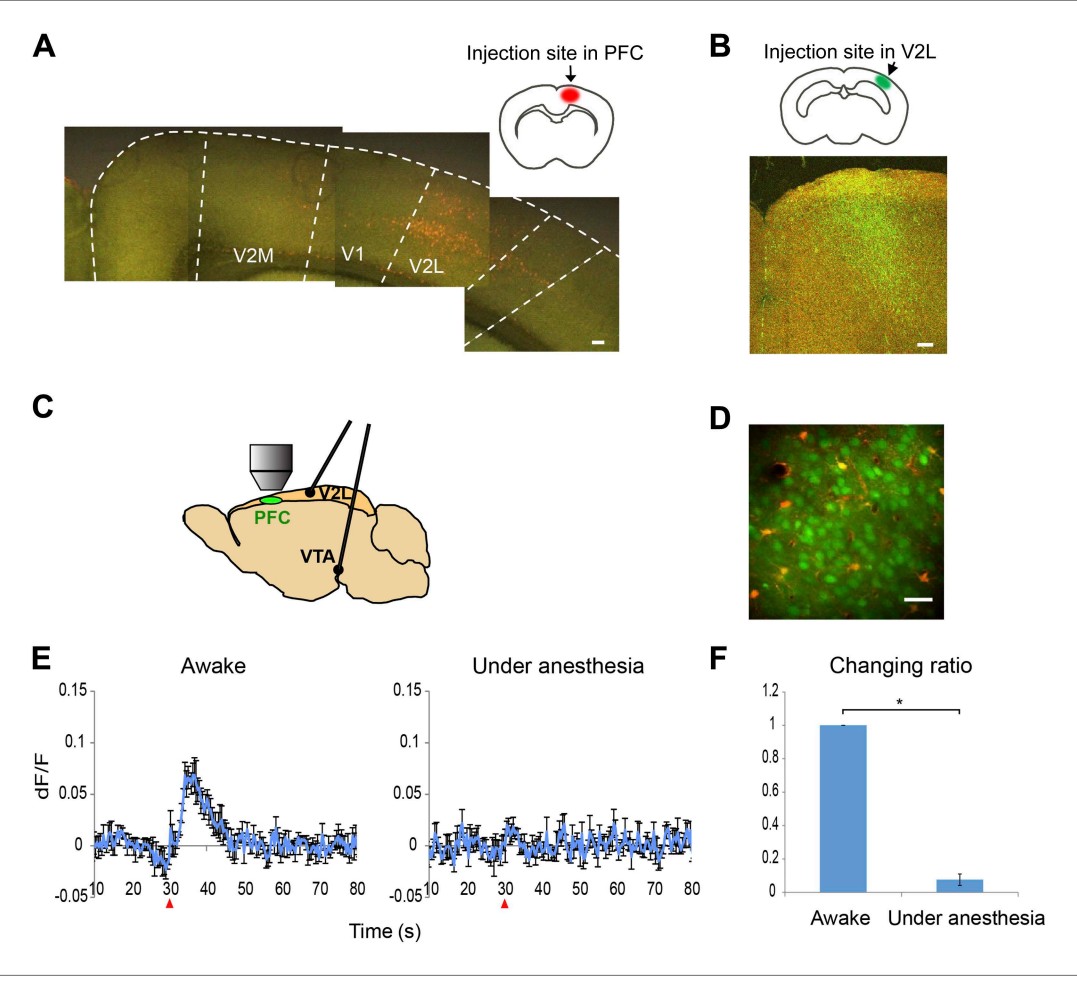

**Figure 1**. Ca²⁺ imaging setup and calcium transients in response to VTA stimulation with and without anesthesia. (**A**) Retrograde tracing. Fluorogold was injected into the prefrontal cortex (PFC). Cells in the secondary visual cortex lateral area (V2L) were labeled. Scale bar, 250 µm. (**B**) Anterograde tracing. FITC-dextran was injected into the V2L. Labeled fibers were observed in the PFC. Scale bar, 100 µm. (**C**) Experimental setup for calcium imaging. A cranial window was opened over M2 (PFC). Two electrodes were implanted in the ventral tegmental area (VTA) and the V2L. (**D**) Two-photon image of calcium indicator (OGB-1)-labeled layer 2/3 cells in M2 (neurons, green; sulforhoda-mine-101 counterstained astrocytes, red to orange). Approximately 50–80 cells were analyzed in each animal. Scale bar, 30 µm. (**E**) The Ca²⁺ transients evoked in an awake animal and in an animal under anesthesia. Population average of Ca²⁺ transients (dF/F) in response to 10 pulses at 50 Hz VTA stimulation (n = 4 animals). VTA electrical stimulation was applied at the 30 s time point (red arrowhead). In contrast to a clear response in awake mice (left), the long-lasting Ca²⁺ transients were not detected in mice under anesthesia (right, 4% isoflurane). The same animals were used for the 'Awake' and 'Under anesthesia' experiments. (**F**) The effect of isoflurane on Ca²⁺ transients. The summed values of Ca²⁺ transients from the 30 to 50 s time points were compared with the 'Awake' value (changing ratio). Paired t test, *p < 0.05. Error bars represent SEM.

administered by intraperitoneal injection (i.p.). Treatment with the D1 antagonist SCH23390 (1 mg/kg) reduced the long-lasting Ca²⁺ transients by 50% and the decay time constant by 60%. In contrast, no detectable attenuation was observed in mice treated with the D2 antagonist eticlopride (0.5 mg/kg) (***Figure 3A,B***). In addition, the short Ca²⁺ response evoked by a 5 Hz VTA stimulation was not affected by the D1 or D2 antagonist (***Figure 3—figure supplement 1A,B***). However, this short Ca²⁺ response was reduced by NMDA and AMPA antagonists (***Figure 3—figure supplement 1C,D***). In the VTA, up to 65% of the neurons are dopaminergic and the others are GABAergic or glutamatergic (***Nair-Roberts et al., 2008***; ***Gorelova et al., 2012***), and some DA neurons co-release glutamate (***Hnasko et al., 2010***). However, glutamatergic terminals are dominant in the mesocortical pathway (***Gorelova et al., 2012***).

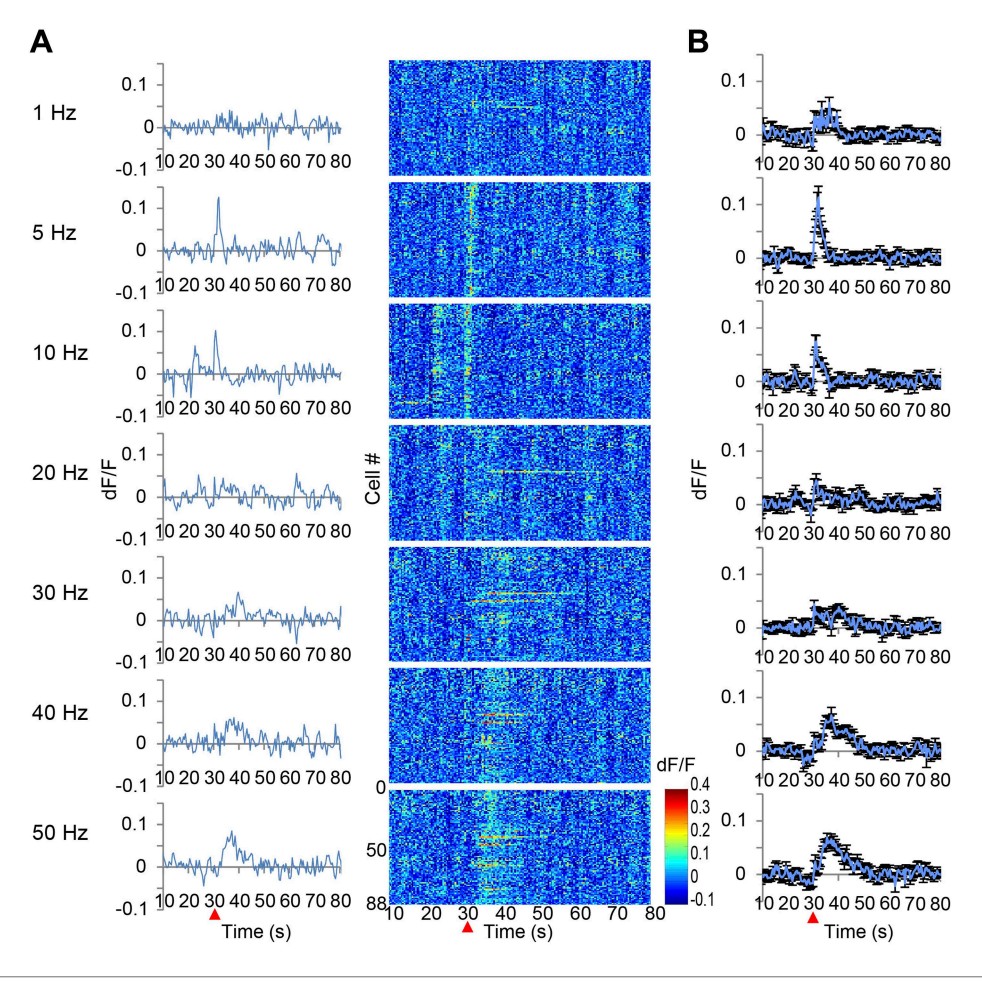

**Figure 2**. Ca²⁺ transients in response to ventral tegmental area (VTA) stimulation. (**A**) The left panel shows the population average of Ca²⁺ transients across the cells of a single animal. The right panel shows the Ca²⁺ transients of each single neuron using a color map according to its dF/F value. A train of 10 pulses was applied at each frequency of VTA stimulation (1, 5, 10, 20, 30, 40, and 50 Hz). (**B**) Population average of Ca²⁺ transients across eight animals. VTA electrical stimulation was applied at the 30 s time point (red arrowhead). Error bars represent SEM.

The following figure supplement is available for figure 2:

**Figure supplement 1**. Ca²⁺ transients in response to secondary visual cortex lateral area (V2L) stimulation.

Inhibition of the short Ca²⁺ response by glutamate receptor antagonists suggests that glutamatergic neurons contribute to the 5 Hz responses. Initial experiments implicated the D1 receptors for the long-lasting Ca²⁺ transient. To exclude a potential role of glutamatergic neurons in initiating the long calcium transients, a cocktail of NMDA and AMPA receptor antagonists was used. However, the long-lasting Ca²⁺ transients were not affected by NMDA and AMPA antagonists (*Figure 3C,D*). This result further suggests that the long-sustained increase in intracellular Ca²⁺ concentration is not mediated by glutamatergic local recurrent networks. Over all, pharmacological experiments clearly suggest D1 receptors are mainly involved in the long-lasting Ca²⁺ response in PFC neurons.

To resolve the potential role of dopamine-induced, long-lasting Ca²⁺ responses for PFC circuitry modification, the PFC neuronal response to sensory input from the V2L after combined repetitive stimulation of the V2L and VTA was investigated. An experimental paradigm composed of a pre-conditioning phase, conditioning phase, and test phase was used (*Figure 4A*). In the pre-conditioning phase, V2L stimulations (20 Hz, five pulses) were given three times to acquire the base response of PFC neurons to V2L inputs. In the conditioning phase, the VTA and V2L were simultaneously

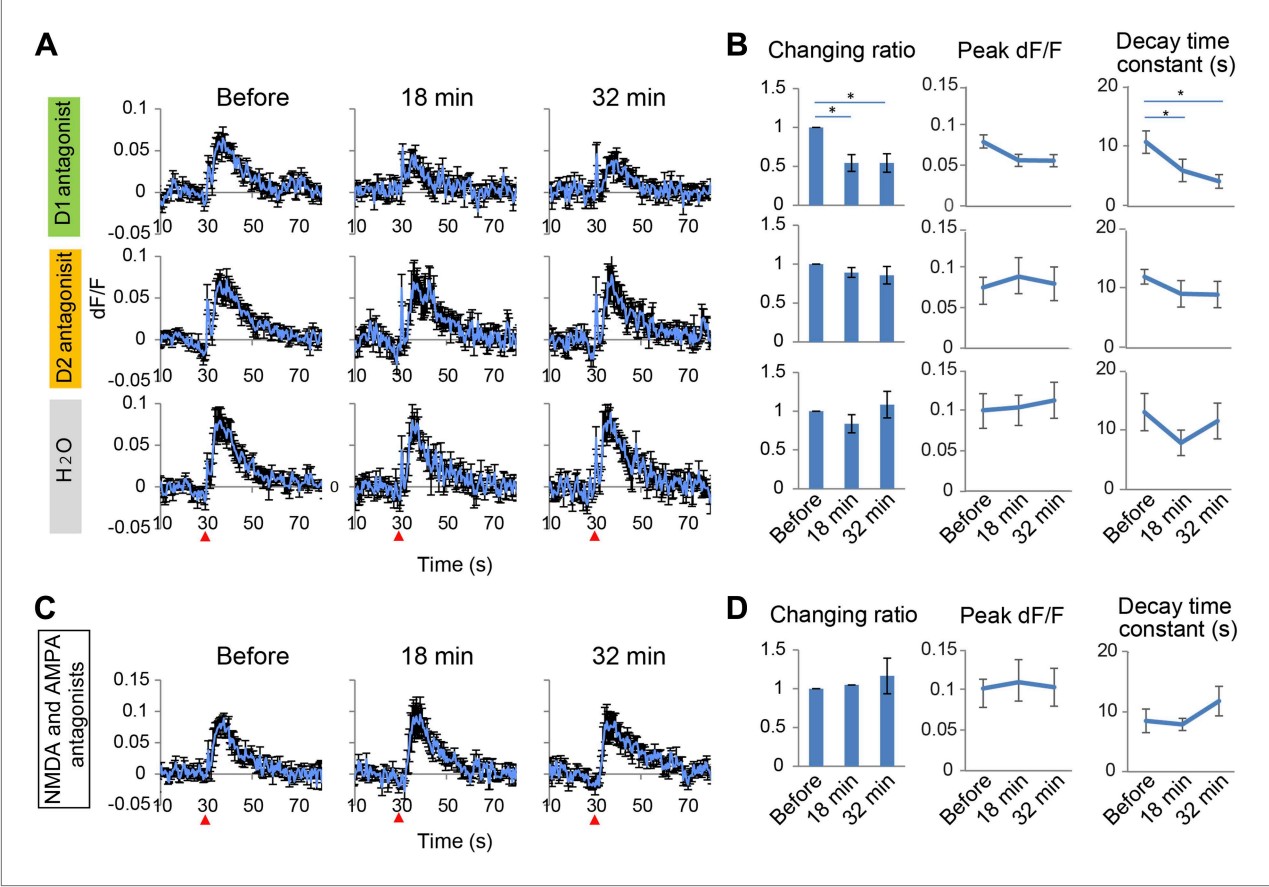

**Figure 3**. Long-lasting $Ca^{2+}$ transients depend on D1 receptors. (**A** and **B**) The effect of the D1 antagonist SCH23390 (1 mg/kg) (upper panel), the D2 antagonist eticlopride (0.5 mg/kg) (middle panel), and $H_2O$, used as a control (bottom panel), on the long-lasting $Ca^{2+}$ transients evoked by 10 pulses at 50 Hz stimulation of the ventral tegmental area (VTA) (n = 6 animals in each experimental group). (**C** and **D**) The effect of a cocktail of NMDA and AMPA antagonists (CPP: 3 mg/kg, and CNQX: 10 mg/kg, i.p.) on long-lasting $Ca^{2+}$ transients (n = 4 animals). (**A** and **C**) Population average of $Ca^{2+}$ transients across animals in each group. VTA electrical stimulation was applied at the 30 s time point (red arrowhead). (**B** and **D**) The changing ratio of the summed values of $Ca^{2+}$ transients between the 30 and 50 s time points compared with the 'Before' value (left panels), the peak of dF/F (middle panels), and the decay time constant (right panels). Paired t test with Holm's adjustment, *p < 0.05. Error bars represent SEM.

The following figure supplement is available for figure 3:

**Figure supplement 1**. Short $Ca^{2+}$ transients do not depend on D1 or D2 receptors but on glutamate receptors.

stimulated every minute for 30 min. After this conditioning phase, repetitive V2L stimulation (20 Hz, five pulses) was applied at three time points: right after, 1 hr after, and 2 hr after completion of the conditioning phase (test phase, *Figure 4A*), to determine whether or not the responses of the PFC neurons were modified.

For the conditioning phase, the effect of timing on V2L and VTA stimulation was also tested by using two different time intervals, Timing1 (T1) and Timing2 (T2) (*Figure 4A*). In T1, V2L stimulation was applied half a second before VTA phasic stimulation. T1 simultaneously simulates a visual experience and VTA activation. For T2, V2L stimulation was applied 45 s after VTA phasic stimulation, which is 15 s before the next VTA stimulation, when no $Ca^{2+}$ transients were observed (*Figure 4A*). Therefore, T2 is used as a control, at a point when visual and VTA inputs are temporally separated. As additional controls, the V2L and the VTA were stimulated separately during the conditioning phase.

Two-way repeated measures ANOVA with temporal change (before, right after, 1 hr after, and 2 hr after conditioning) and conditioning paradigm (V2L only, VTA only, Timing1, and Timing2) as factors, revealed that population averages of dF/F values of $Ca^{2+}$ transients signal showed significant differences in temporal change ($F(3, 84) = 48.486$, p < 0.0001) and the conditioning paradigm × temporal

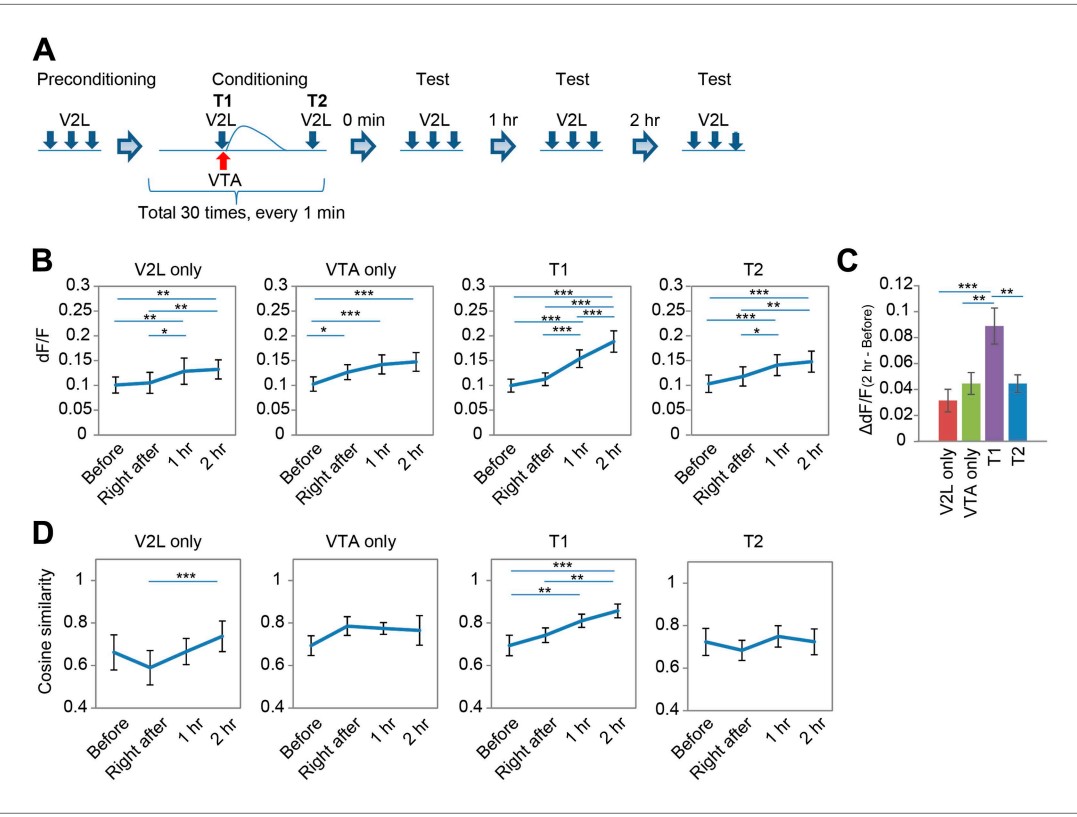

**Figure 4**. Combined repetitive stimulation of the secondary visual cortex lateral area (V2L) and the ventral tegmental area (VTA) causes a modification of prefrontal cortex (PFC) neuronal response. (**A**) Experimental design for stimulation of the V2L and VTA composed of three phases: pre-conditioning, conditioning, and test. Five pulses at 5 Hz and 10–15 pulses at 50 Hz were used for V2L and VTA electrical stimulation, respectively. T1: Timing1; T2: Timing2. (**B**) Shift in population average of dF/F value. Post hoc tests revealed that dF/F values were significantly different in many of the comparisons between the time points in each conditioning paradigm (Ryan's test). (**C**) Changing dF/F value calculated by subtracting the 'Before' value from the '2 hr after' value (dF/F$_{(2\ hr-before)}$) to simplify the results of (**B**). The T1 conditioning paradigm showed a significantly larger temporal change than the others. (**D**) Difference in pattern similarity. Pattern similarity calculated from cosine similarity revealed that only T1 conditioning significantly increased the value of '2 hr after' when compared with the 'Before' value. n = 8 animals in each experimental group. Ryan's post hoc test: *p < 0.05, **p < 0.01, *p < 0.001. Error bars represent SEM.

The following figure supplements are available for figure 4:

**Figure supplement 1**. Percentage distributions of neurons by dF/F in each conditioning group and reliability of calcium transients occurrence following three repetitive secondary visual cortex lateral area (V2L) stimulations.

**Figure supplement 2**. No correlation between Ca$^{2+}$ influx evoked by ventral tegmental area (VTA) stimulation and neuronal activity increase in response to secondary visual cortex lateral area (V2L) stimulation.

change interaction ($F(9,84) = 3.173$, $p = 0.0024$) (*Figure 4B*). T1 conditioning significantly increased the dF/F value more than other conditioning paradigms, indicating that T1 conditioning has the largest modification effect on the PFC response to a sensory input (one-way ANOVA: $F(3,28) = 6.504$, $p = 0.0018$; post hoc Ryan's test: $p < 0.01$; *Figure 4C*). This increase in value is due to an increased number of neurons showing high dF/F values (*Figure 4—figure supplement 1A*). In addition, there was no significant correlation between the Ca$^{2+}$ level in response to VTA stimulation (at conditioning) and the increased Ca$^{2+}$ level in response to V2L stimulation '2 hr after' T1 conditioning (test phase) (*Figure 4—figure supplement 2*), suggesting that the increased neuronal response with V2L stimulation is independent of how well the neuron responds to the DA input. This relatively unexpected

conclusion may be explained by the fact that the network dynamics of the PFC microcircuit are composed of inhibitory and excitatory neurons, both of which express DA receptors.

Finally, the neuronal population dynamics were investigated through analysis of the pattern similarity (cosine similarity, see the 'Materials and methods' section) of neuronal activity in the PFC across three repetitions of V2L stimuli, conducted at four time points: 'before', 'right after', '1 hr after', and '2 hr after' after the conditioning phase. Two-way repeated measures ANOVA on the temporal changes of the mean cosine similarities in each conditioning paradigm revealed significant differences in the temporal factor ($F(3, 84) = 7.676$, $p = 0.0001$) and the time × condition paradigm interaction ($F(9, 84) = 2.483$, $p = 0.0145$; *Figure 4D*). Post hoc tests also showed that the cosine similarity in T1 was significantly increased in '2 hr after' compared with 'before' (Ryan's test, $p < 0.01$; *Figure 4D*). This increase in cosine similarity in T1 conditioning is not simply due to an increased number of high dF/F neurons (*Figure 4—figure supplement 1A*), because '1 hr after' in T1 conditioning had a significantly increased cosine similarity (*Figure 4D*) but its population probability distribution was not significantly different from that of VTA only (2 hr after) or T2 (2 hr after) (two-sample Kolmogorov–Smirnov test, $p = 0.5128$ and $p = 0.1418$, respectively) (*Figure 4—figure supplement 1A*). This suggests that the increase in cosine similarity in T1 conditioning was not due to an increased number of high dF/F neurons. In addition, the increased cosine similarity was not due to increased reliability of $Ca^{2+}$ transients occurrence in response to three repetitive V2L stimulations, because the reliability calculated by Cronbach's alpha did not change between 'before' and '2 hr after' in any of the conditioning groups, including T1 (*Figure 4—figure supplement 1B*). These results indicate that only T1 conditioning improved the pattern similarity in PFC neurons, and the increase in pattern similarity could be the result of circuit (network) modification, and not simply the result of increased reliability of $Ca^{2+}$ transients occurrence or an increased number of high responding dF/F neurons.

This study revealed that only coincident visual sensory input with long-lasting $Ca^{2+}$ transients increased neuronal activity and induced robust repeatable neuronal responses, at the population level, to the same sensory input. By increasing both dF/F and pattern similarity, DA input may enhance PFC activity and establish cortical representation (*Xue et al., 2010*). In fact, it is known that DA release plays a major role in dynamic cortical remodeling in the auditory cortex (*Bao et al., 2001*), suggesting that the phenomenon observed here in the T1 conditioning paradigm may represent the neurophysiological basis for dynamic neural events. Besides network modification in PFC, it has also been demonstrated that PFC neurons show long-lasting $Ca^{2+}$ transients that depend on the D1 receptor pathway. This newly identified physiological phenomenon might have an important function in $Ca^{2+}$ regulation of neuronal processes (*Berridge, 1998*; *Brenowitz et al., 2006*; *Roussel et al., 2006*; *Bardo et al., 2006*). For example, association learning is only successful if the cue signal is less than 20–30 s before the reward signal. Long-lasting $Ca^{2+}$ transients may be tightly associated with this time window in association learning. Taken together, the long-lasting $Ca^{2+}$ transients reported here could be a key phenomenon required to explain the dynamics of the dopaminergic neural network and its role in PFC cognitive functions.

## Materials and methods

### Neuronal tracing

Neuronal retrograde tracer Fluorogold (4%; Fluorochrome, Denver, CO) or FITC-conjugated dextran (Life Technologies, Carlsbad, CA) were injected into the M2 or V2L, respectively, and 48 hr later, mice were fixed with 4% formalin in phosphate buffer. Brains were sliced using a vibratome and Fluorogold or FITC was identified by using fluorescence microscopy.

### Animal preparation

All procedures were conducted according to the animal welfare guidelines of the NIH and approved by the NIH Animal Care and Use Committee.

C57BL/6 mice ranging in age from 2 to 4 months were used. Throughout all procedures, body temperature was maintained at 37°C using a heating pad. Anesthesia was induced with Avertin (2.2.2-tribromoethanol; Sigma-Aldrich, St. Louis, MO), and mice were placed in a stereotaxic device.

Stainless bipolar stimulating electrodes were implanted in the V2L (2.5 mm lateral and 2.5 mm posterior from bregma, 0.5 mm below the cortical surface) and VTA (0.5 mm lateral and 3.1 mm posterior from bregma, 4.5 mm below the cortical surface).

A 1.5 mm craniotomy (with the dura carefully removed) was opened over M2 (0.5 mm lateral and 1.0 mm anterior from bregma).

Multi-cell bolus loading of neocortical cells with the calcium indicator Oregon-BAPTA Green 1-AM (OGB-1-AM; Life Technologies) and astrocyte marker sulforhodamine 101 (SR101; Life Technologies) was performed as previously described (*Stosiek et al., 2003*; *Nimmerjahn et al., 2004*). This multi-cell bolus loading was performed in superficial layer 2/3 (L2/3). The craniotomy was then covered with silicone (Kwik-Sil Adhesive; World Precision Instruments, Sarasota, FL) and sealed with a glass cover-slip. A metal bar was glued directly on the skull with dental acrylic for future attachment to an imaging frame. About ~2 hr after surgery, when the mouse had completely recovered from the anesthesia, cortical activity, measured by evoked $Ca^{2+}$ transients, was imaged in the awake, head-fixed mouse.

## Two-photon imaging

The mouse was placed under the microscope. Cortical activity was imaged using a two-photon microscope (Olympus Fluoview; Olympus, Japan) equipped with a 25 × (1.05 NA) water-immersion objective (Olympus). Excitation wavelength was 870 nm (Mai-Tai oscillator; Spectra-Physics, Santa Clara, CA). Images (256 × 256 pixels) were acquired at a frame rate of 2.3 Hz. Low quality images (when cells showed unclear boarders) were not used for analysis.

Imaging started at time 0, however high background signals (~0.03 dF/F, lasting 2–3 s) were present at time 0 due to mechanical noise. Therefore, figures show the $Ca^{2+}$ transients starting from 10 s to demonstrate the neuronal activity that occurs in response to electrical stimulation (*Figures 1–3*, *Figure 2—figure supplement 1* and *Figure 3—figure supplement 1*).

## Drug administration

SCH23390 (1 mg/kg), eticlopride (0.5 mg /kg), CPP (3 mg/kg), and CNQX (10 mg/kg) (Sigma–Aldrich) were administered by i.p. injection. The injected volume was adjusted to 1% of the animal's body weight.

## Microstimulation protocols

For VTA stimulation, a biphasic pulse of 1 ms duration was used in all experiments. To evoke the long-lasting $Ca^{2+}$ transients, 50–400 µA and 10–15 pulses were applied. The experimental protocol shown in *Figures 1–3* used 400 µA and 10 pulses. VTA stimulation with 50–400 µA and 10–15 pulses was used for the conditioning experiment (*Figure 4*). For the V2L, a biphasic pulse of 1 ms and 250 µs duration was used in the experiments shown in *Figure 2—figure supplement 1* and *Figure 4*, respectively. Stimulation with 350 or 400 µA was used in *Figure 2—figure supplement 1*. To evoke dF/F values below 0.2, in response to V2L stimulation at 'before', current was adjusted (50–300 µA) in each animal (experiments shown in *Figure 4*).

## Data analysis

Data were analyzed with custom-written programs in MATLAB (Mathworks, Natick, MA) and ImageJ (NIH, Bethesda, MD). To remove motion artifacts from in vivo calcium imaging, the ImageJ plugin TurboReg for image alignment (*Thévenaz et al., 1998*) was used. Individual cells were semi-automatically detected (*Supplementary file 1*). SR101-stained astrocytes were excluded from data analysis. For each cell, fluorescence change was defined as dF/F = (F1 − F0)/F0, where F1 is fluorescence at any time point, and F0 is the baseline fluorescence, defined as the median of fluorescence values measured within 40 s before and after the time point of F1 (for responses to VTA stimulation), or within 2.5 s before and after the time point of F1 (for responses to V2L stimulation). To calculate the decay time constant (tau) of $Ca^{2+}$ transients (*Figure 3B,D*), the following equation was used for fitting the time course of dF/F during the decay period: dF/F(t) = $dF/F_{max} \times e^{-t/tau}$, where $dF/F_{max}$ is the peak dF/F value and t is elapsed time after dF/F peaks. To calculate the population average of dF/F shown in *Figure 4*, the averages across cells from the three V2L stimulations were averaged. Pattern similarity in *Figure 4D* was measured using cosine similarity (cosine of the angle between two vectors) in each pair of vectors, which are composed of the N-dimension of dF/F values of each neuronal response, where N is the total number of neurons. For the three repetitions of V2L stimulation at each time point, three cosine similarities were measured between the first and second, second and third, and first and third V2L stimulations, and then these three cosine similarities were averaged to show the pattern similarity of the time points. In *Figure 4B*, the result of post hoc tests revealed that dF/F values were significantly different in many of the comparisons between the time points in each conditioning paradigm (Ryan's test, $p < 0.05$). In *Figure 4C*, to simplify, the values of dF/F at 2 hr after conditioning were

compared, and standardized by subtracting the 'before' values from '2 hr after' among the different conditioning groups (dF/F$_{(2\text{ hr}-\text{before})}$).

## Acknowledgements

I thank K. Wang for providing the research environment required to perform these experiments and for helpful discussions, G. Seabold, E. Tytell, and M. Yoshizawa for critically reading the manuscript, G. Ashida for helpful discussions, and K. Nakao and R. Renden for additional scientific comments on the manuscript.

## Additional information

### Funding

| Funder | Author |
| --- | --- |
| National Institute of Mental Health | Motoko Iwashita |
| Japan Society for the Promotion of Science | Motoko Iwashita |

The funders had no role in study design, data collection and interpretation, or the decision to submit the work for publication.

### Author contributions

MI, Conception and design, Acquisition of data, Analysis and interpretation of data, Drafting or revising the article

### Ethics

Animal experimentation: All procedures were conducted according to the animal welfare guidelines of the NIH and approved by the NIH Animal Care and Use Committee (#GCP-01-07).

## Additional files

### Supplementary file

• Supplementary file 1. ImageJ macros for detecting individual cells.

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
