## [Decision Letter]

Thank you for sending your work entitled “Phasic activation of VTA neurons increases response and pattern similarity in PFC neurons” for consideration at *eLife.* Your article has been favorably evaluated by Senior editor, a Reviewing editor, and 2 reviewers.

The following individuals responsible for the peer review of your submission have agreed to reveal their identity: Michael Hausser (Reviewing editor) and Jeremy Seamans (one of two peer reviewers).

The Reviewing editor and the other reviewers discussed their comments before we reached this decision, and the Reviewing editor has assembled the following comments to help you prepare a revised submission.

In this manuscript, the author has investigated the contribution of dopamine to the modification of neuronal representations in the PFC (M2) by using in vivo two-photon Ca^2+^ imaging techniques in awake mice. The author reported long-lasting (∼30sec) calcium transients in the PFC evoked by high-frequency electrical stimulation of the VTA, which depends on D1 receptors. In addition, the author showed that simultaneous activation of the VTA and V2L can cause long-term (∼2h) increases of calcium transients in the PFC to V2L inputs. This study is interesting and novel both in its approach and findings, and should make a significant contribution to the field. Several issues need to be addressed before publication.

Major comments:

1) Figure 4: Analysis of pattern similarity differences. Before analyzing the pattern similarity of neuronal population activity, the author needs to check whether the increase of the population averages of dF/F values from 'Before' to '2h' with T1 conditioning (Figure 4) is due to the increase of the numbers of the neurons activated by V2L stimulation, or due to the increase of the activity level of the neurons by V2L stimulation without changing the numbers of activated neurons. If the numbers of activated neurons by V2L stimulation was drastically changed, cosine similarity might not be a fair method for comparing the similarity of activity patterns across different conditions.

2) “Importantly, these long-lasting Ca^2+^ transients were only detected in awake mice, but not in anesthetized ones (Figure 1)...” This is a very interesting result, but also puzzling. If the mechanism of long-lasting calcium transients is due to D1 as the author showed, this effect should be observed also in the anesthetized state. It would be helpful if the author could speculate about the reason why the long-lasting calcium transients are state-dependent. Please also confirm whether the results in left ('Awake') and right ('under anesthesia') panels of Figure 1 were obtained from the same mice or the different ones.

3) There are serious reservations about the IP3 data. First, because the effect is so strong compared to the other manipulations and second because (by necessity) 2-APB was applied so differently from the other drugs. It was the only drug applied to the cortical surface (versus i.p.). Plus it was analyzed at the whole field level. Given these differences, either the data should be excluded from the manuscript, or more experiments should be done to confirm these conclusions.

4) In Figure 1 the author tries to make the point that the effect is seen in awake and not anesthetized mice. This needs to be quantified by group statistics.

5) Figure 3 describes the effects of DA antagonists on the Ca transients. The author needs to do a more thorough job of characterizing what is changing (peak height, decay, duration etc.) and what is not. Same for glutamate and IP3 mediated effects.

6) In Figure 4, observed increase in cosine similarity could simply be result from increased reliability of occurrence of calcium transients, not from increased population pattern similarity. It would be beneficial to analyse reliability of calcium transients in each experimental epochs.

7) The author states that 50-80 cells were analyzed per animal. Were these treated independently and the same way as cells from other animals? More importantly, what is the variance in the effect across the population?

8) “VTA neurons are mainly dopaminergic”: this description sounds incorrect. Dopaminergic neurons are up to 65% in VTA, and the rest are GABAergic and glutamatergic (Cohen JY et al Nature 2012; [6]; [16]; etc.). The strong glutamatergic nature of the pathway is perfectly consistent with the effects of glutamate blockers on the transient responses (Figure 3—figure supplement 1). The author should discuss the effects of non-dopaminergic afferents to M2 on the results reported in this paper.

9) Some portion of M2 pyramidal neurons have projections to the VTA (Watabe-Uchida M et al., Neuron. 2012). Has the author considered or ruled out the possibility of antidromic stimulation of M2 when using VTA electric stimulation?

10) In Figure 3 was water really used as a control?

[Editors' note: further revisions were requested prior to acceptance, as described below.]

Thank you for resubmitting your work entitled ”Phasic activation of ventral tegmental neurons increases response and pattern similarity in prefrontal neurons.” for further consideration at *eLife.* Your revised article has been favorably evaluated by a Senior editor, a Reviewing editor, and the original two reviewers. The manuscript has been improved but there are some remaining issues that need to be addressed before acceptance, as outlined below:

From the Reviewing editor:

The title should state “prefrontal cortex neurons" rather than just ”prefrontal neurons".

The Abstract needs some rewording: e.g. “By establishing a system of in vivo two-photon Ca^2+^ imaging in awake mice” change to ”Using in vivo two-photon population Ca^2+^ imaging in awake mice”; “Phasic stimulation of dopaminergic neurons in the ventral tegmental area (VTA) evoked long-lasting (∼30 sec) calcium transients in PFC” - in PFC neurons (specify layer or cell type if possible); "Pharmacological analysis revealed that this long-lasting calcium transients are regulated by dopamine D1 receptor-dependent pathway." Please clarify: the long-lasting enhancement of the transients, or the transients themselves?; “Furthermore, only visual sensory input applied 0.5 second before the VTA dopaminergic input could evoke higher Ca^2+^ transients”; what does 'higher' mean?).

Figure 1: Ensure that the contrast to the line indicating the average is high enough. Clean up the axis labelling: why start from 10 s and not 0 s. The SI unit is “s” and not “sec”. A 0-line is nice, but doesn't replace an x-axis.

Figure 1: Drop the grid lines; the significance line should end centered on the data bars.

Reviewer #1:

The author has done a good job with the revisions. I would still suggest the following:

The text reads: ”In VTA, up to 65% of the neurons are dopaminergic and the others are GABAergic or glutamatergic (19)(20), and some DA neurons co-release glutamate (21)novel,, suggesting that glutamatergic/DA neurons might also contribute to the 5Hz responses.” Two points here. First, overall this is true for the VTA but the mesocortical projection which is what is being studied, is mainly glutamatergic not DAergic (See papers by Morales or Gorelova). Second, the results actually show that glutamate and not DA contributes the 5Hz response. Therefore, I cannot understand the equivocation when a firm conclusion can be drawn.

In the text and Figure 3–figure supplement 2: Once again I do not agree with including this in the paper. Not only for the reasons I mentioned in the last round but also because of the simple fact that just because IP3 and DA manipulations both decrease the Ca response, it in no way means they are related. To imply this given the existing data is not wise in my opinion.

In terms of H20 control I'm surprised that a hypo-osmotic solution didn't affect the cells and the Ca signal?

The document should be proofread to check that tenses are correct etc.

Reviewer #2:

1) Abstract. The sentence “Furthermore, ....” is a bit confusing, because “only visual sensory input applied 0.5 second before the VTA dopaminergic input ” is the protocol during the conditioning, and “evoke higher Ca^2+^ transients and increase pattern similarity” is the effect 2 hours after the conditioning, but this timeline (or the experimental causality) is not clearly described.

2) Figure 1. It would be helpful if the author can provide the picture or sketch of the injection site of the tracer as an insertion panel.

3) Figure 1. “Population average of Ca^2+^ transients (dF/F) in response to a 50-Hz-VTA”. Please provide the information of the duration or numbers of the pulses of the electrical stimulation, in the figure legend or main text. The author describes it in the Method section, but it is better to have the information earlier in the manuscript.

---

## [Author Response]

*1) Figure 4: Analysis of pattern similarity differences. Before analyzing the pattern similarity of neuronal population activity, the author needs to check whether the increase of the population averages of dF/F values from 'Before' to '2h' with T1 conditioning (Figure 4) is due to the increase of the numbers of the neurons activated by V2L stimulation, or due to the increase of the activity level of the neurons by V2L stimulation without changing the numbers of activated neurons. If the numbers of activated neurons by V2L stimulation was drastically changed, cosine similarity might not be a fair method for comparing the similarity of activity patterns across different conditions*.

Thank you for your suggestion. To address this comment, I have compared the percentage distributions of neurons pooled from all animals in the same group (V2L only, VTA only, T1 and T2; Figure 4—figure supplement 1). The distribution shifted to a higher dF/F after 2h later in all groups (Figure 4—figure supplement 1; comparing ‘before’ and ‘2h’) and the largest shift was observed in T1 compared to other conditioning groups. To test whether the higher cosine similarity observed in T1 was only due to the increased number of high responsive cells, I focused on the data from T1 (1h after), VTA only (2h after) and T2 (2h after). Among these results, only T1 (1h after) had a significantly increased cosine similarity (Figure 4), but its population probability distribution was not significantly different from that of VTA only (2h after) and T2 (2h after) (Two sample Kolmogorov-Smirnov test p=0.5128 and p=0.1418, respectively) (Figure 4—figure supplement 1), suggesting that the increased cosine similarity in T1 is not only due to an increased number of high responsive cells. I also describe these points in more detail in the Results section.

*2) “Importantly, these long-lasting Ca*^*2+*^
*transients were only detected in awake mice, but not in anesthetized ones (Figure 1)...” This is a very interesting result, but also puzzling. If the mechanism of long-lasting calcium transients is due to D1 as the author showed, this effect should be observed also in the anesthetized state. It would be helpful if the author could speculate about the reason why the long-lasting calcium transients are state-dependent. Please also confirm whether the results in left ('Awake') and right ('under anesthesia') panels of Figure 1 were obtained from the same mice or the different ones*.

Isoflurane inhibits multiple voltage-dependent calcium channels (25)(9). Because I used a high dosage of isoflurane (4%), there is a possibility that the calcium current might be blocked. I described this point in the main text.

Thank you for the question in the last sentence. The same mice were used and I clarified this information in the figure legend for Figure 1.

3) There are serious reservations about the IP3 data. First, because the effect is so strong compared to the other manipulations and second because (by necessity) 2-APB was applied so differently from the other drugs. It was the only drug applied to the cortical surface (versus i.p.). Plus it was analyzed at the whole field level. Given these differences, either the data should be excluded from the manuscript, or more experiments should be done to confirm these conclusions.

As the reviewers mentioned above, the drug administration procedure for 2-APB was different from the other drugs. Initially I tried i.p. injection with the concentration used for experiments, and ended up focusing on other organs rather than the brain, because I could not detect an inhibitory effect of 2-APB in Ca^2+^ response in the brain. This suggests the drug does not cross the blood-brain barrier. As stated in the manuscript, the analysis was done according to whole field recordings, not at the single cell level, causing an inconsistency in experimental procedure. However, I would like to make the argument that it is important to consider a mechanism for the long-lasting calcium transients, and this experiment provides a potential source for the change seen with them. To address the reviewers’ concerns, I have moved the figure to the supplemental section and I described the procedure for this experiment more clearly in the main text of the Results and Material and methods sections.

4) In Figure 1 the author tries to make the point that the effect is seen in awake and not anesthetized mice. This needs to be quantified by group statistics.

I have added the statistical analysis recommended by the reviewers (Figure 1).

5) Figure 3 describes the effects of DA antagonists on the Ca transients. The author needs to do a more thorough job of characterizing what is changing (peak height, decay, duration etc.) and what is not. Same for glutamate and IP3 mediated effects.

Thank you for the above suggestions. I have added a more detailed description for the peak dF/F value and the decay (Figure 3).

To further characterize the decay, the decay time constant (tau) was calculated by fitting the time course of dF/F during the decaying period using the following equation:

dF/F(t) = dF/F_max_ * e ^-t/tau^, where dF/F_max_ is the peak dF/F value and t is the elapsed time after dF/F reached its peak. This is described in the Materials and methods section.

6) In Figure 4, observed increase in cosine similarity could simply be result from increased reliability of occurrence of calcium transients, not from increased population pattern similarity. It would be beneficial to analyse reliability of calcium transients in each experimental epochs.

In response to the reviewers’ comment, I analyzed the reliability of the calcium transient occurrence across three repetitive V2L stimulations by calculating Cronbach’s alpha (Figure 4—figure supplement 1). In all groups, Cronbach’s alpha of ‘Before’ and ‘2h’ are not significantly different, suggesting that the increase in cosine similarity may be caused by a change at the network level; not by an increased reliability of calcium transient occurrence. I added this analysis and results in the main text and a figure (Figure 4—figure supplement 1).

7) The author states that 50-80 cells were analyzed per animal. Were these treated independently and the same way as cells from other animals? More importantly, what is the variance in the effect across the population?

Cells were treated independently and analyzed in the same way as cells from other animals. For an example, to test whether the difference in the number of cells analyzed per animal changes the result, I analyzed the correlation between the increase in population average of dF/F and the number of cells analyzed in each animal from the T1 conditioning group. The regression analysis showed no significant correlation between them (adjusted R^2^ = -0.1144, standardized regression coefficient = -0.212, P = 0.614). So, I think as a whole, the number of cells analyzed per animal does not have major effect on the analysis across the animals.

8) “VTA neurons are mainly dopaminergic”: this description sounds incorrect. Dopaminergic neurons are up to 65% in VTA, and the rest are GABAergic and glutamatergic (Cohen JY et al Nature 2012; [6]; [16]; etc.). The strong glutamatergic nature of the pathway is perfectly consistent with the effects of glutamate blockers on the transient responses (Figure 3—figure supplement 1). The author should discuss the effects of non-dopaminergic afferents to M2 on the results reported in this paper.

Thank you for this comment. I have added points to address this observation in the main text.

9) Some portion of M2 pyramidal neurons have projections to the VTA (Watabe-Uchida M et al., Neuron. 2012). Has the author considered or ruled out the possibility of antidromic stimulation of M2 when using VTA electric stimulation?

I did not rule out antidromic effects on M2 neurons, however, long lasting Ca^2+^ transients and short Ca^2+^ transients were reduced by D1 and glutamate blocker, respectively, suggesting that these responses are mainly evoked by synaptic transmission.

10) In Figure 3 was water really used as a control?

I used H_2_O as a control because I diluted drugs for i.p. injection with H_2_O. The injected volume was adjusted to 1% of the animal’s body weight.

[Editors' note: further revisions were requested prior to acceptance, as described below.]

The title should state “prefrontal cortex neurons” rather than just “prefrontal neurons”.

Thanks for your comment. I corrected the title as suggested.

*The Abstract needs some rewording: e.g. “By establishing a system of in vivo two-photon Ca*^*2+*^
*imaging in awake mice” change to “Using in vivo two-photon population Ca*^*2+*^
*imaging in awake mice”; “Phasic stimulation of dopaminergic neurons in the ventral tegmental area (VTA) evoked long-lasting (∼30 sec) calcium transients in PFC” - in PFC neurons (specify layer or cell type if possible); “Pharmacological analysis revealed that this long-lasting calcium transients are regulated by dopamine D1 receptor-dependent pathway.” Please clarify: the long-lasting enhancement of the transients, or the transients themselves?; “Furthermore, only visual sensory input applied 0.5 second before the VTA dopaminergic input could evoke higher Ca*^*2+*^
*transients”; what does 'higher' mean?).*

I reworded the Abstract according to the reviewer’s comments and tried to maintain the word limit (150 words).

Figure 1: Ensure that the contrast to the line indicating the average is high enough. Clean up the axis labelling: why start from 10 s and not 0 s. The SI unit is “s” and not “sec”. A 0-line is nice, but doesn't replace an x-axis.

I corrected the line colors and width to ensure high contrast throughout the figures.

Concerning the reviewer’s comment about the x-axis, while the recordings were started at time 0, the first few seconds of recording showed high background signals (∼ 0.03 dF/F, lasting for 2-3 sec) which was due to mechanical noise. Therefore, I showed the calcium transients starting from 10s to only show the neuronal activity that occurred in response to the electrical stimulation. I have now included this information in the Material and methods.

Figure 1: Drop the grid lines; the significance line should end centered on the data bars.

Thanks, I made the recommended changes.

Reviewer #1:

The author has done a good job with the revisions. I would still suggest the following:

The text reads: “In VTA, up to 65% of the neurons are dopaminergic and the others are GABAergic or glutamatergic (19)(20), and some DA neurons co-release glutamate (21)novel,, suggesting that glutamatergic/DA neurons might also contribute to the 5Hz responses.” Two points here. First, overall this is true for the VTA but the mesocortical projection which is what is being studied, is mainly glutamatergic not DAergic (See papers by Morales or Gorelova). Second, the results actually show that glutamate and not DA contributes the 5Hz response. Therefore, I cannot understand the equivocation when a firm conclusion can be drawn.

Thank you for the comments. I rewrote the sentence to strengthen the conclusion that the 5Hz response is mainly due to glutamatergic neurons.

In the text and Figure 3–figure supplement 2: Once again I do not agree with including this in the paper. Not only for the reasons I mentioned in the last round but also because of the simple fact that just because IP3 and DA manipulations both decrease the Ca response, it in no way means they are related. To imply this given the existing data is not wise in my opinion.

Considering the reviewer’s concerns, I removed the section about the 2-APB experiments.

In terms of H20 control I'm surprised that a hypo-osmotic solution didn't affect the cells and the Ca signal?

Thanks for the observation. However, I feel that my experiments show that the H_2_O does not affect the Ca^2+^ responses. Therefore, using H_2_O injections that are1% of the animal’s body weight has no noticeable effect.

The document should be proofread to check that tenses are correct etc.

I have had multiple readers check the document, and I hope to have made all of the appropriate corrections.

Reviewer #2:

*1) Abstract. The sentence “Furthermore, ....” is a bit confusing, because “only visual sensory input applied 0.5 second before the VTA dopaminergic input ” is the protocol during the conditioning, and “evoke higher Ca*^*2+*^
*transients and increase pattern similarity” is the effect 2 hours after the conditioning, but this timeline (or the experimental causality) is not clearly described.*

I rewrote sentence more clearly to address the reviewer’s concern.

2) Figure 1. It would be helpful if the author can provide the picture or sketch of the injection site of the tracer as an insertion panel.

I added diagrams of the two injection sites in Figure 1.

*3) Figure 1. “Population average of Ca*^*2+*^
*transients (dF/F) in response to a 50-Hz-VTA”. Please provide the information of the duration or numbers of the pulses of the electrical stimulation, in the figure legend or main text. The author describes it in the Method section, but it is better to have the information earlier in the manuscript.*

Thanks, this is a valid point. I added the stimulation protocol to the Figure legends to clarify the procedure used.